# Results of the MOVE MS Program: A Feasibility Study on Group Exercise for Individuals with Multiple Sclerosis

**DOI:** 10.3390/ijerph20166567

**Published:** 2023-08-12

**Authors:** Brynn Adamson, Nic Wyatt, Latashia Key, Carrena Boone, Robert W. Motl

**Affiliations:** 1Department of Health Sciences, University of Colorado Colorado Springs, 1420 Austin Bluffs Pkwy, Colorado Springs, CO 80907, USA; 2Department of Kinesiology and Community Health, University of Illinois at Urbana-Champaign, 506 S. Wright St., Urbana, IL 61801, USA; 3Department of Recreation, Sport, and Tourism, University of Illinois at Urbana-Champaign, 506 S. Wright St., Urbana, IL 61801, USA; 4Department of Kinesiology and Nutrition, University of Illinois Chicago, 1200 West Harrison St., Chicago, IL 60607, USA

**Keywords:** multiple sclerosis, exercise, disabled persons, feasibility study

## Abstract

Exercise improves a wide range of symptoms experienced by those living with multiple sclerosis (MS) and may foster community and a positive sense of disability identity. However, exercise rates remain low. Sustained exercise participation has the greatest likelihood of improving symptoms and requires a theory-based approach accounting for the barriers faced by people with MS that impede exercise participation long-term. MOVE MS is a once weekly group exercise program based on Social Cognitive Theory supporting long-term exercise participation through peer instruction, behavior change education, multiple exercise modalities, and seated instruction. This feasibility study evaluated MOVE MS with a 7-month trial. The primary scientific outcome was exercise participation and the secondary outcomes were MS symptoms/impact, self-efficacy, depression, anxiety, disability identity, and quality of life, among others. We further conducted semi-structured formative interviews post-intervention. Thirty-three participants began the program. The onset of COVID-19 necessitated a shift toward online delivery. Seventeen participants completed the program. There were non-significant improvements in exercise participation (Godin Leisure-Time Exercise Questionnaire, baseline mean = 14.2 (SD = 11.8), post-intervention mean = 16.6 (SD = 11.2), F-value = 0.53 (Partial Eta^2^ = 0.08), and several secondary outcomes (including the MS Impact Scale, MS Walking Scale, and the Leeds MS Quality of Life Scale). Sixteen participants were interviewed, and analysis yielded five themes on program components and feedback. MOVE MS—delivered in-person or online—may be a feasible option for long-term exercise programming for people with MS.

## 1. Introduction

Exercise is a promising behavior for multiple sclerosis (MS) management. Over 50 trials indicate that exercise can improve the physiological and psychological consequences of MS including fatigue, balance, mobility, strength, depression, and the related outcomes such as quality of life [1,2,3,4,5]. Exercise is associated with reduced occurrence of relapse and other positive neurological outcomes [4]. Group exercise, in particular, has been associated with improvements in balance [6,7], fatigue [8], fall-risk [7], mobility [7], symptom improvement [9,10], and quality of life [8,11], and may be an empowering way to manage MS within a supportive environment [12]. Nevertheless, exercise participation rates remain low among those with MS [13]. This may be explained by barriers identified through qualitative research among those with MS, including environmental, social, and personal barriers [14], and one recent study identified environmental barriers as particularly relevant for long-term change [15]. A key priority set by a working group and the National Multiple Sclerosis Society is developing effective, evidence-based, theoretically driven physical activity interventions that can increase physical activity behavior among persons with MS [16].

Very little is known about long-term exercise behavior change and its benefits for those living with MS. However, among other neurological populations, positive long-term exercise behavior change has been demonstrated. This includes those with Parkinson’s disease [17,18,19,20], the traumatic brain injury population [21], post-stroke populations [22,23], and those living with spinal cord injury [24]. Furthermore, positive exercise behavior change, and health benefits have been demonstrated with these long-term programs in community-based [17,18,20,22,25] and group settings [17,18,20,22].

This collectively highlights a need for building long-term exercise programs for the MS population that (1) acknowledge the unique barriers for sustained exercise engagement and (2) are initially evaluated for feasibility.

Importantly, the salient barriers must be contextualized in an understanding of disability and the ways in which those with MS experience disability and illness. Recent trends towards using exercise as “medicine” may promote exercise in ways that reinforce personal responsibility for health and ignore socially-imposed, disabling barriers [26,27,28] Such approaches can be psychologically damaging for those faced with structural exercise barriers and who experience disability episodically and/or invisibly [29]. By promoting exercise to avoid disability and framing disability as essentially negative, exercise becomes a means of self-discipline and source of shame and guilt, instead of a resource for well-being [30]. Given the strong evidence for the physical, psychological, and quality of life benefits associated with exercise, it is vital to ensure that exercise programs are theoretically driven, evidence-based, focused on long-term behavior change, and consider health communication messaging that is ethically sound and uses disability-inclusive language. It is essential to consider the importance of inclusive language in exercise promotion in this population in order to best support sustainable behavior change without the negative psychological side effects of internalized ableism and shame which have been demonstrated to negatively impact the process of disability identity navigation [28,31].

Bogart has demonstrated that disability identity—or a positive sense of self and community related to having a disability—is protective of both depression and anxiety in the MS population [32]. However, persons with MS are not likely to report a positive sense of disability identity, partially due to MS being acquired later in life [33,34] and partially due to the episodic and often invisible nature of living with MS [29]. Furthermore, it is unclear how to support disability identity navigation in this community while also addressing physical inactivity. MOVE MS is an ongoing group exercise program for individuals with MS that promotes exercise as a resource and helps individuals navigate a positive sense of disability identity through autonomy, confidence, creativity, and social support. MOVE MS promotes sustained exercise participation based on a multi-level approach consistent with Social Cognitive Theory (SCT) [35]—a widely used theory for prompting exercise behavior change among persons with MS [36]. MOVE MS has been iteratively delivered in community-based recreation facilities in Central IL since 2018 and runs weekly. It, therefore, represents a long-term program which has been sustainable in a center-based model.

MOVE MS uniquely targets many salient and reported environmental, social, and personal barriers to exercise for those with MS. The MOVE MS program partners with accessible facilities, demonstrates adapted exercise modalities (first from a seated position prior to demonstrating from a standing position), emphasizes an MS-specific community program, allows for personalization of at-home exercise participation, addresses conflicting exercise information, provides many modality options for exercise and emphasizes choice, utilizes respectful and empowering language in class [28,37,38,39,40], and focuses on a positive sense of disability identity with the language used by instructors in the class (e.g., this exercise does not serve to fix your MS but as a resource for overall well-being) [41,42]. Through incorporating principles of SCT, MOVE MS addresses many of the above-mentioned barriers by incorporating peers as experts [43,44], emphasizing personal autonomy and confidence-building, and providing a supportive environment to help individuals navigate a positive sense of disability identity through exercise.

Before determining the efficacy of the MOVE MS program for changing exercise behavior long-term, we first sought to implement and evaluate a shorter-term trial of MOVE MS with a feasibility study design [45]. The specific aims of this project were: (1) to determine the feasibility and needs associated with delivering MOVE MS in preparation for an efficacy trial, and (2) to determine the initial efficacy of the MOVE MS program for increasing exercise participation as well as secondary health and psychological outcomes.

Some key uncertainties regarding the implementation of a long-term exercise program among those with MS are, firstly, whether the design of the MOVE MS program reduces participant burden and contributes to long-term adherence. Secondly, this study seeks to identify the extent to which a peer-delivered group exercise program can impact disability identity navigation. Therefore, this study seeks to determine whether the once per week dose is sufficient to impact changes in PA behavior and manageable for participants to incorporate into their lives long-term. Prior to determining the efficacy of the MOVE MS program for changing exercise behavior long-term and impacting psychological outcomes including disability identity, we sought to implement and evaluate a shorter-term trial of MOVE MS with a feasibility study design [45].

## 2. Materials and Methods

### 2.1. Program Design

MOVE MS is a 3-part program delivered in successive units within a community setting, and the complete graphical illustration of the program timeline is illustrated in Figure 1. We designed a 7-month program held once per week for 1 h for this pilot feasibility study, although it is intended for implementation on an ongoing basis.

#### Program Description

Part 1 Jumpstart—a 4-session introduction to exercising with MS. Classes, 60 min long, were held once per week. The session involved behavior change strategies informed by SCT. The sessions involved pre-class questions, active discussion between class members, key definitions, and a written component (e.g., writing down your SMART goal, writing down your biggest exercise barriers, etc.) The following topics were covered: Session 1: Outcome Expectations, Session 2: Goal Setting and Self-Monitoring, Session 3: Self-Efficacy and Relapse, and Session 4: Barriers and Facilitators. Key definitions and recommendations were introduced during these sessions such as the MS Physical Activity Guidelines, differences between exercise and lifestyle physical activity, and examples of these that could be completed between classes. The classes provided social support, education, and exposure to the following types of exercise: Yoga (adaptive Iyengar style), functional exercise (balance, stretching, resistance training), Pilates (postural control, balance, breathing, flexibility), and Zumba^®^ Gold (aerobic Latin-inspired dance-based exercise).

Part 2 Discover—once-per-week exercise classes were held for 60 min, divided into four 6-week modules covering: Pilates, Yoga, functional exercise, and Zumba^®^ Gold.

Part 3 Boost—The purpose of the Boost sessions was four-fold: (1) to refresh concepts from Jumpstart, i.e., goal-setting check-ins, building confidence, etc., (2) to rekindle camaraderie, (3) to reinforce what was learned in the previous module with at-home instructions, and (4) to review the topic of the next Discover module.

All classes were delivered by certified instructors, some of whom had MS (peer-delivery). Pilates and functional exercise modules were peer-delivered (certified by The Neuro Studio^®^), the Zumba^®^ module was delivered by the PI (licensed in Zumba^®^ Gold), and the yoga module was delivered by licensed instructors who have previously worked with populations living with disability. The PI and/or research assistants were present in all classes for safety and support.

### 2.2. Research Design

This project adopted a pre/post mixed methods feasibility design. The research was approved by the University of Illinois at Urbana–Champaign Institutional Review Board (Protocol #18121). Feasibility metrics, scientific quantitative outcomes, and qualitative surveys and interviews were collected to evaluate the program. The purpose of conducting a mixed methods design at this stage was to gather subjective program experience information and triangulate with survey data and feasibility data [46].

#### 2.2.1. Recruitment

We recruited through local MS support groups, news outlets, MS events, and host centers. Interested participants were screened for the following inclusion/exclusion criteria: (1) 18+ years of age, (2) diagnosis of MS, (3) relapse-free in the past 30 days, (4) willing and able to participate in an exercise program, and (5) one or fewer affirmatives on the Physical Activity Readiness Questionnaire (those with two affirmatives could participate with a physician’s clearance and those with two+ affirmatives were excluded for safety reasons). Level of MS impairment was not an inclusion/exclusion criterion.

#### 2.2.2. Research Procedures

Participants who passed screening were mailed a copy of the Informed Consent Document, and a questionnaire packet of the Scientific Outcomes (presented below). Participants were then given a set of equipment (yoga blocks, yoga strap, TheraBand) and a program manual. After completion of Jumpstart and before Discover, participants completed the questionnaire packet again. Participants were invited to complete a feedback survey regarding the Jumpstart program and each of the four Discover modules. After program completion, participants completed the questionnaires a third time. All participants, including those who dropped out, were invited to engage in a semi-structured interview about personal experiences and feedback for the program’s improvement. Participants were paid USD 10 for each set of completed questionnaires and for completing the interview for a total of USD 40.

#### 2.2.3. Feasibility Outcome Measures

We captured metrics of feasibility including Process, Resource, Management and Scientific outcomes as conducted in previous exercise programs in the MS population [38,39,40]. Process metrics focused on critical program processes (e.g., recruitment, enrollment, retention). Resource metrics focused on time (research team time interfacing with participants, data collection and entry time, etc.) and resource needs (i.e., instructional materials, postage, exercise equipment, participant remuneration, instructor costs, facility costs). Management metrics focused on data management needs and intervention delivery fidelity. Lastly, Scientific metrics focused on scientific efficacy, participant burden, and participant safety.

#### 2.2.4. Qualitative Data Collection

Consistent with recommendations from O’Cathain et al. [47], a qualitative interview was developed to enhance key components of the feasibility study. Key feasibility information needed from the qualitative interview included: gaining insight into the context of intervention delivery, intervention acceptability (as well as the acceptability of different components of the intervention), perceived value, perceived benefits, perceived harm or unintended intervention consequences, process evaluation insights (e.g., recruitment method acceptability, adherence, dose, evaluation burden), and learning what changes participants would have liked to have seen in the intervention design and delivery. Based on these needs, a semi-structured interview guide was developed (see Appendix A. The decision was made to conduct one-on-one interviews rather than complete focus groups to gather individualized information and ensure dissenting views from the group were not masked [47].

Following intervention completion, semi-structured interviews were held on Zoom (version 5.7.7; Zoom Video Communications, San Jose, CA, USA) or over the phone with the PI (female, PhD, 7+ years of experience in qualitative research, adaptive group exercise instructor) and research assistant (non-binary, BS, 2+ years of experience in qualitative research, identifies as disabled). There were consistent interactions between the PI, research assistant, and the participants prior to engaging in the semi-structured interviews (communication about the study, and during classes). Interviews were recorded and transcribed for analysis.

#### 2.2.5. Scientific Outcome Measures

We utilized the following questionnaires to collect information about participant characteristics: a demographic questionnaire (participant descriptors), and the Patient-Determined Disease Steps (PDDS) (participant descriptors of MS impairment) [48]. To measure scientific feasibility we utilized the 4-item Godin Leisure-Time Exercise Questionnaire [49,50] (GLTEQ) as the primary outcome measure, and the following as secondary outcome measures: Multiple Sclerosis Impact Scale (MSIS), which measures the physical and psychological impact of MS on a person’s life [51]; Multiple Sclerosis Walking Scale (MSWS-12), which measures a person’s perceived walking ability [52]; Hospital Anxiety and Depression Scale (HADS), which is a validated tool for assessing symptoms of anxiety and depression in clinical populations [53]; University of California Los Angeles Loneliness Scale (UCLALS), which assesses a person’s subjective experience of loneliness [54]; Life Orientation Test—Revised (LOT-R), which assesses expectations for future outcomes [55]; the Leeds MS Quality of Life Questionnaire (LMSQOL), which is an MS-specific measure of quality of life [56]; the Exercise Self-Efficacy Scale (ESES), which assesses a participant’s confidence to engage in exercise across a variety of situations [57]; The Exercise Motivations Inventory-2 (EMI-2), which assesses 14 dimensions of exercise motivation [58]; the Identity Reconstruction Assessment Scale (IRAS), which assesses identity reconstruction among those living with MS [59]; and the University of Washington Self-Efficacy Scale for People with Disabilities and Chronic Conditions Short-Form (UWSES), which assesses disability management self-efficacy among those living with chronic health conditions [59].

### 2.3. Statistical Analysis

We calculated recruitment, retention, adherence, compliance, and attrition rates per cohort, and compared pre/post scores on the GLTEQ and secondary outcome measures using an ANOVA across timepoints (T1, T2, and T3). All data were analyzed in SPSS Statistics, Version 28 (IBM Corporation, Armonk, NY, USA); we provide means, standard deviations, F-values, and effect sizes (Partial Eta^2^) for all outcome measures between T1 and T2, as well as between all timepoints. Data were analyzed for all persons who completed the study (completer’s analysis). Missing data items were substituted using a process of item imputation where the answer from the preceding timepoint was carried forward to the missing item (last observation carried forward).

### 2.4. Qualitative Analysis

We adopted a pragmatic epistemology consistent with incorporating qualitative interviews in mixed methods feasibility studies [46,47] and utilized an inductive phenomenological thematic analysis method. The goal of using qualitative research in addition to the quantitative and feasibility metrics was to obtain complementarity and triangulation support [46]. The intent of the interview analysis was to understand the experiences participants had while engaging in MOVE MS and gather feedback for future program iterations. All interviews were listened to, read, and coded by at least two team members with four research team members participating in the analysis. Two team members were frequently involved with the participants during the intervention and the other two were not. Team members who were and were not involved with the participants during the intervention were matched during the coding process to allow for an “insider” and “outsider” perspective in the coding. A codebook was developed after open coding three interviews and discussing the open codes from four coders. The codebook was continually revised throughout the coding of the remaining 13 interviews. Themes identified represent patterns identified as domain summaries of feedback provided by participants [60]. Qualitative methods and reporting below utilized the Consolidated Criteria for Reporting Qualitative Research (COREQ) checklist criteria [61].

#### Quality

To ensure quality in our analysis, we followed guidelines in accordance with phenomenological analysis, namely bracketing and psychological reduction [62], ensuring themes were internally consistent and supported by ample participant quotations [63,64]. Therefore, throughout the qualitative analysis, authors LK and CB provided challenging and questioning comments to refine the themes and their descriptions. Furthermore, as recommended by Smith and McGannon, choosing quality criteria consistent with our epistemological and methodological frameworks was more important than a set of universal qualitative research criteria for qualitative research [65].

## 3. Results

### 3.1. Recruitment and Enrollment

The research team contacted 188 potential participants (individuals from past research studies or who have previously expressed interest in research participation) via email and sent recruitment materials (fliers) to the support group leaders in: Bloomington/Normal, IL, USA; Champaign/Urbana, IL, USA; Peoria, IL, USA; and Colorado Springs, CO, USA (where the PI had moved to a new institution). Figure 2 outlines the flow of participants through the various parts of the program.

### 3.2. Baseline Participant Demographics

The participants were mainly female (75.8%), white (87.9%), and the mean age was 55.2 years (SD = 1.7). The majority had relapsing remitting multiple sclerosis (RRMS) (78.1) and 54.5% had a patient determined disease steps (PDDS) of 2 or below. The participant baseline demographic information is in Table 1.

### 3.3. Intervention Delivery

Two cohorts were enrolled for in-person participation at two partner facilities in central IL, USA (Bloomington and Urbana, IL, USA) in February 2020. The MOVE MS program was halted in March 2020 after the fourth session (the Jumpstart program completed) due to the COVID-19 pandemic. After 4 months with no resolution, we resumed the class in an online format. Of the 16 participants who had not dropped out during or prior to the Jumpstart program (weeks 1–4 of the active intervention), only 4 participated in the remote version of the program. The reasons for non-participation included: health reasons = 4, too busy with work from home/childcare responsibilities = 3, uninterested in remote exercise = 1, Zoom difficulties = 1, and no response = 3. Based on the high drop-out rate, two additional cohorts were recruited to begin a fully online program. These two cohorts began in January 2021. All participants received a safety session and Zoom tutorial prior to resuming or beginning MOVE MS online. This was not originally planned for the MOVE MS program. A cohort-by-cohort flow through the program is outlined in Figure 3.

We sent a new baseline packet (T2.5) to the four participants who consented to participate in the new remote program (along with a new informed consent document outlining that the program would now be held via Zoom), and four participants returned their packets. All participants who resumed the program were from cohort 1 (none from cohort 2). Another participant dropped out during the first Discover module because of difficulties using Zoom on a cell phone (no computer available); they could not see the instructor well enough to follow.

An important challenge with the shift to a remote format was ensuring safety. We therefore created a safety session and Zoom tutorial session to be held before the program resumed for Cohort 1 and to be held as the first official MOVE MS session for cohorts 3 and 4. This covered important safety guidelines, including discussion of the rating of perceived exertion scale so participants could monitor their exertion at home during class and maintain moderate or light intensity, discussion of preparing a safe and hazard-free exercise space in their homes, the selection of a sturdy and suitable chair to sit in, and an introduction to the equipment used. Additionally, this safety session served as a time to troubleshoot Zoom and familiarize participants with the program and controls. All participants received a packet with all safety and Zoom troubleshooting information.

### 3.4. Feasibility Metrics

The complete results of the feasibility metrics are presented in Table 2.

### 3.5. Scientific Feasibility

Due to large loss throughout the program, only 14 participants completed outcome questionnaires across all timepoints and, therefore, these results should be interpreted with caution. The differences between timepoints across primary and secondary outcomes were normally distributed. No outcomes showed significant improvement between timepoints. However, Exercise Self-Efficacy was significantly reduced between T1 and T2, but not T2 and T3. Notably, on a few packets, participants wrote notes in the margins stating that answers were changed by the stress of the pandemic. Appendix B: Table A1 presents the treatment outcomes at baseline (T1), post-Jumpstart program (T2), and at follow-up (T3).

### 3.6. Program Feedback

Feedback on the program was gathered at five timepoints (after the Jumpstart session and after each Discover module). The mean scores are presented in Appendix C: Table A2. Participant feedback on the program was overwhelmingly positive.

### 3.7. Qualitative Results from Interviews

Sixteen participants opted to participate in the interview following completion of the program (two of these participants dropped out, the other fourteen completed the whole program). Our inductive thematic analysis resulted in several themes related to participant experiences within MOVE MS.

#### 3.7.1. Facilitators and Barriers to MOVE MS Participation

Interestingly, COVID-19 was brought up by very few as a barrier to participation (who either started in person and had to switch to the online format or who simply preferred in-person exercise) but, overwhelmingly, participants named having the class offered virtually as one of the biggest facilitators to participation, even if they discussed some technical difficulties. Virtual participation removed the potential for fatigue from driving to the center or walking from the parking lot to the fitness studio and preserved energy for the actual exercise class. Participant 400 was part of the in-person cohort that transitioned to the online format and described the benefits of an online program:


*I would say the biggest difficulty was just my own technical difficulty, but honestly, having the Zoom class that was so convenient because for me, like fatigue is one of my big issues, and it takes a lot of energy for me to get up and get ready, get dressed, get out of the house to the health center. And then that’s a pretty good walk. And then, exercise for an hour and then go home. With the zoom, you can just go to the chair so you can save all that energy for the class itself. Honestly, the Zoom classes were kind of a godsend.*


Other facilitators included having the social support of the group (discussed further in the theme Group Dynamic), and the adaptability of the different exercises (including seated demonstrations with standing options).

Important barriers to participation included: low self-confidence and health-related barriers which were both MS and non-MS related. Environmental barriers came up a few times in relation to finding an exercise space at home with a wall and where they could use their computer. Most mentioned that they were able to find a suitable location but the home environment as a barrier to home exercise should be noted.

#### 3.7.2. Module and Instructor Feedback

Participants appreciated having multiple exercise modalities as well as multiple instructors. Most participants could identify a favorite and least favorite module (which varied evenly across the four modules), but all mentioned that they were glad they tried all of the modules. The participants were generally highly satisfied with the duration of the class sessions (60 min) as well as the length of the modules (8 weeks, once per week). In terms of intensity, there was more variability as some participants who were younger and/or had less MS impairment discussed wanting more intense exercises, while those with more MS impairment were generally satisfied with the module intensity.

Participants appreciated having a peer instructor with MS. They appreciated hearing the peer instructor discuss varying challenges affecting her ability to exercise in specific ways and that this provided encouragement and confidence that they could adapt in the presence of symptoms or challenges.


*I think definitely having an instructor with MS, was super helpful. Just because it gives you like that extra confidence of like, well she can do it I can do it. It was all made it easier to kind of see how her body would react and be like, oh okay so that is normal… So for her to offer those suggestions based on the way her own body was responding was very helpful to me.*
(Participant 203)

A few participants stated that, while it was an added benefit to have a peer instructor with MS, as long as the instructor was knowledgeable about MS, that is what mattered most to them. Therefore, training for future MOVE MS instructors should incorporate extensive knowledge regarding MS.

#### 3.7.3. Group Dynamic

Overwhelmingly, the most important component of the program was the group dynamic. Participants discussed that their desire for community was a large motivating factor for choosing to participate in the MOVE MS program. The group members provided support, accountability, and encouragement.


*Before class, I [wasn’t sure] if I could keep up with these people. But, obviously, it doesn’t matter. We’re all going through different things… After probably the first month, we kind of formed that team. And we’re all there for each other, which was good, and actually very supportive.*
(Participant 307)

Several stated that the single most important factor that helped them stick with the program was the camaraderie of their group:


*Knowing the people that I’ve been online with get excited when I logged in and they’re all on and they’re all talking and stuff like that. So, I think that’s what got me motivated to—to stay on with the program.*
(Participant 304)

When asked about whether MOVE MS impacted their understanding of disability, many were unsure how to answer but reaffirmed that participating with a peer instructor and with a group of other peers with MS was a large contributing factor to why they kept coming and why they hoped the program would continue.

#### 3.7.4. Suggestions for Improvement

There were three key suggestions for improving the program based on the interviews as well as on the written open-ended feedback. Firstly, participants expressed the desire to have recorded videos to refer to between sessions so that they could exercise more. Though they were given pictures in their manuals, these did not provide the level of support that they felt they needed to be able to successfully exercise between sessions.


*I wish there was a bank of videos to watch on the days we didn’t have class because I was definitely not as good about [exercising] when we didn’t have a scheduled class, but I felt like if I could put in my calendar, I’d watch the video and do it. That’s just how I am. If it’s not an actual obligation and I can put a video on or like have someone show me how to do it, then it’s hard for me to find that time.*
(Participant 203)

The second key suggestion was having options for different times to meet for class in case of a conflict. Lastly, participants suggested providing more set-up and conversation time before the hour of exercise started. Participants who started the program in person did miss the in-person interaction and suggested that we start class a bit early to replicate the in-person socializing that would naturally occur before and after the in-person classes.

#### 3.7.5. MOVE MS is a Reminder That MS Affects Everyone Differently

When asked if their participation in MOVE MS impacted their understanding of disability, many stated that it did not “per se”, however, the majority described the ways that participating in a group of others with MS reminded them of the vast variability of MS impairment.


*I don’t know that [MOVE MS] changed [my understanding of disability], but I’ve been reminded that people with MS come in all shapes and sizes and colors and levels of impairment. It’s interesting that there are some symptoms that almost everyone identifies with and others that are particular to that individual.*
(Participant 108)

Another participant described the ways that MOVE MS provided her with a more complete understanding of disability through getting to know other participants:


*In the general sense of disability, I would say no, however, you know you have a piece of pie that represents disability, right? So, what MOVE MS did was, okay, I have my pie disability but now I have a little bit more information to add to the picture, so I might add a little whipped cream and a little whatever on top of the pie because it’s just an additional information to complete the picture that much more. Just interacting with people that are in the same boat, you know, just hearing what they might be met with or working through that day, or, you know, what their life is like. Yeah, I just think it made the picture a bit more complete.*
(Participant 204)

## 4. Discussion

This preliminary study of the MOVE MS program determined the feasibility of a group exercise program designed for long-term involvement among people with mild, moderate, and severe impairment due to MS. Though the feasibility of implementing this program as originally designed could not be ascertained, due to the shift from in-person to online classes brought about by the COVID-19 pandemic, our team determined the process, resource, management, and scientific feasibility of an online group exercise class for individuals with all levels of MS impairment.

Regarding process feasibility, attrition rates were extremely high in the first two cohorts (ultimately, 2 completed and 18 began, giving 11% retention), but attrition was much more closely aligned with other exercise studies in the MS population [66,67,68,69] with cohorts 3 and 4 who were recruited to a fully online program 1 year after the beginning of the pandemic (14 completed and 17 began, giving 82.4% retention). Our eligibility rate was very high (96.8%) based on the inclusion of persons with mild, moderate, and severe MS within the program. Attendance in class sessions provided our adherence metric which ranged between 67% in cohort 1 and 86% in cohort 3, demonstrating the feasibility of a once per week exercise class for persons with any level of MS impairment. The qualitative data support that the group camaraderie was a major factor leading to high attendance in the online cohorts.

Regarding resource feasibility, we determined the communication needs, staff training time, and monetary costs for an online group-based intervention and an in-person intervention, since costs were initially incurred and training did occur. Due to the overwhelming preference for the online intervention among those recruited in cohorts 3 and 4, as well as the reduced monetary resources needed to implement an online program, this will be further evaluated as a sustainable long-term program. Furthermore, there may be additional benefits to maintaining an online group exercise program such as the accessibility for those struggling with fatigue, the convenience for those who live farther away from a center or who work (both mentioned by participants), and potentially increasing motivation across those with different physical skill levels, which has been demonstrated among older adults [70].

Regarding management feasibility, staff time needs were ascertained. Additionally, we determined that, among the completed questionnaire packets, there was a low rate of missing data items (ranging between 99.4% and 100% complete). However, we were missing several full data packets at each timepoint (even among those who continued with the program). Future trials should incorporate digital data collection measures to reduce participant burden (e.g., taking packets to the post office) and reduce staff time to collect data (1674 min for questionnaire packet preparation and processing) without sacrificing validity [66]. Furthermore, future evaluations of MOVE MS will reduce the number of survey items to reduce the possibility of survey fatigue.

Regarding scientific feasibility, only one adverse event was reported (which may have been unrelated to the program). The event was resolved, and the participant continued with the program through to its completion. Therefore, MOVE MS, as a seated, home-based exercise program, is viewed as safe for individuals with MS. There were no significant treatment effects, as MOVE MS did not significantly increase exercise participation as measured by the GLTEQ. One key finding was a significant reduction between T1 and T2 in Exercise Self-Efficacy Scale scores. This was likely driven by the in-person cohorts who completed both timepoints and were responding to the impact of the pandemic. Future MOVE MS trials will involve self-reflection of changes in Self-Efficacy to identify whether participants truly feel less confident in their ability to exercise in the near future. We will further focus on the most desirable outcomes, as the extensive survey battery might have yielded survey fatigue and obfuscated the capturing of program benefits.

The focus on the group component (community building through the Jumpstart program and maintenance of group cohesion throughout) was acknowledged to be the most important benefit of MOVE MS according to the qualitative interviews despite a lack of significant improvements in loneliness according to the UCLALS. Including the Social Provisions Scale [71] and the Physical Activity Group Environment Questionnaire [72] in future trials of MOVE MS may provide insight into the impact of the camaraderie and group cohesion. This indicates the potentially influential role of disability community in supporting adherence to the program as well as self-reported mood benefits from participation. Qualitative feedback from the program indicated that the exercise group was an essential factor in their adherence to the program as well as having an impact on their understanding of MS.

### 4.1. Limitations

As the program began immediately prior to the beginning of the COVID-19 pandemic, the research approach had to be adapted. We offered MOVE MS through the video conferencing software program, Zoom. This helped to overcome other barriers to participation (including transportation needs/distance from the center, and fatigue from travel to the center), but presented new barriers (e.g., discomfort using Zoom, not having a suitable space at home to exercise, loss of tactile feedback from exercise instructors to assist participants). The later cohorts had much greater comfort in using Zoom due to the widespread usage over the course of 2020. By the time the intervention started for cohorts 3 and 4, almost all had extensive experience with Zoom or another video conferencing software.

Perhaps the most consistent challenge was the collection of data packets. There were mail delays with the USPS throughout the latter half of 2020, and some packets were lost in the mail and had to be replaced. Some participants assured the team that the packets would be sent soon but then they were not after long periods of time elapsed. An important finding related to process feasibility is the need for data collection to be digital (e.g., REDCap or Qualtrics) to allow for quick reminders and bypassing mail delays and problems.

Lastly, there was a wide range of disability present in the sample. This likely contributed to the variance in outcome measures at each timepoint. Given the large standard deviations across outcomes, it is likely that the variance presented challenges in detecting changes in outcomes. However, a great strength of this program is that it is one of the first to provide all-inclusive, integrated group exercise across the spectrum of MS impairment. In future studies, a larger sample will be recruited in order to conduct analyses based on MS impairment group (i.e., mild, moderate, and severe).

### 4.2. Future Directions

In addition to increasing sample size to allow for possible variance in effects based on impairment level, there are several areas for future development within this program: firstly, identifying and implementing clear strategies for increasing physical activity behavior outside of group exercise classes. This may be through additional jumpstart sessions focused on lifestyle physical activity behavior, wearable self-monitoring, or greater focus on using the logbook; and secondly, creating a repository of exercise videos for use outside of class time to increase likelihood of exercise participation between classes. Thirdly, given that this is an on-going community-based program, another important future direction is to increase the length of the trial to capture adherence rates and exercise participation rates for one year+ of participation.

One last important area for future study is to compare the current outcomes with those of a replicated trial outside of the context of the pandemic. Comparing preference for a virtually delivered exercise intervention now that in-person activities have resumed could provide insight into the ways in which virtually delivered home-based exercise interventions can overcome barriers to exercise participation for this population.

## 5. Conclusions

Given the challenges and the need to significantly alter the structure of the MOVE MS intervention, an additional pilot trial built on this feasibility study is needed to determine the treatment efficacy of changing exercise behavior long-term and the impacts on physical and psychosocial outcomes.

## Figures and Tables

**Figure 1 ijerph-20-06567-f001:**
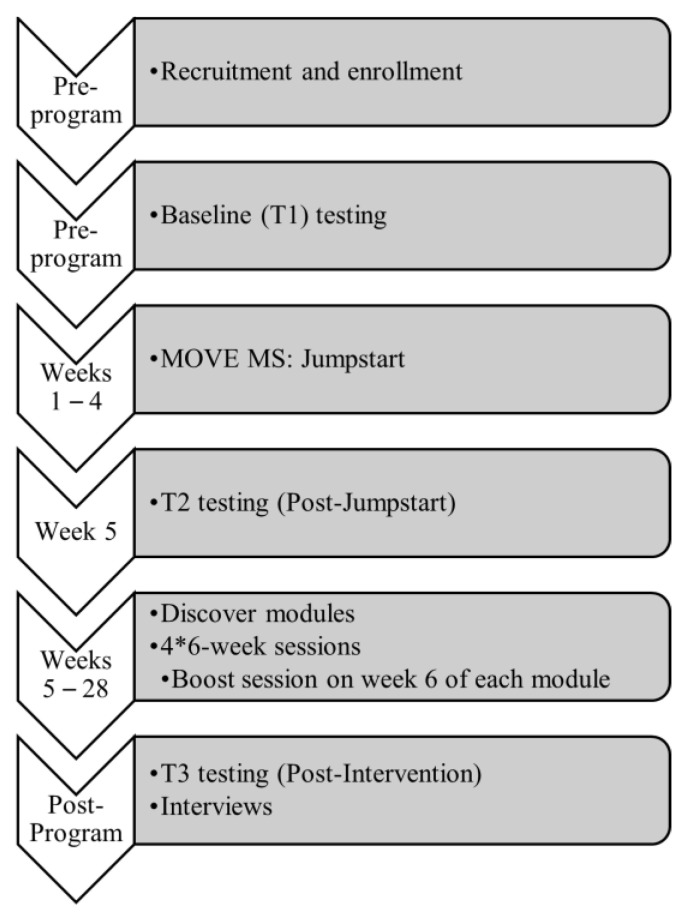
Program Timeline.

**Figure 2 ijerph-20-06567-f002:**
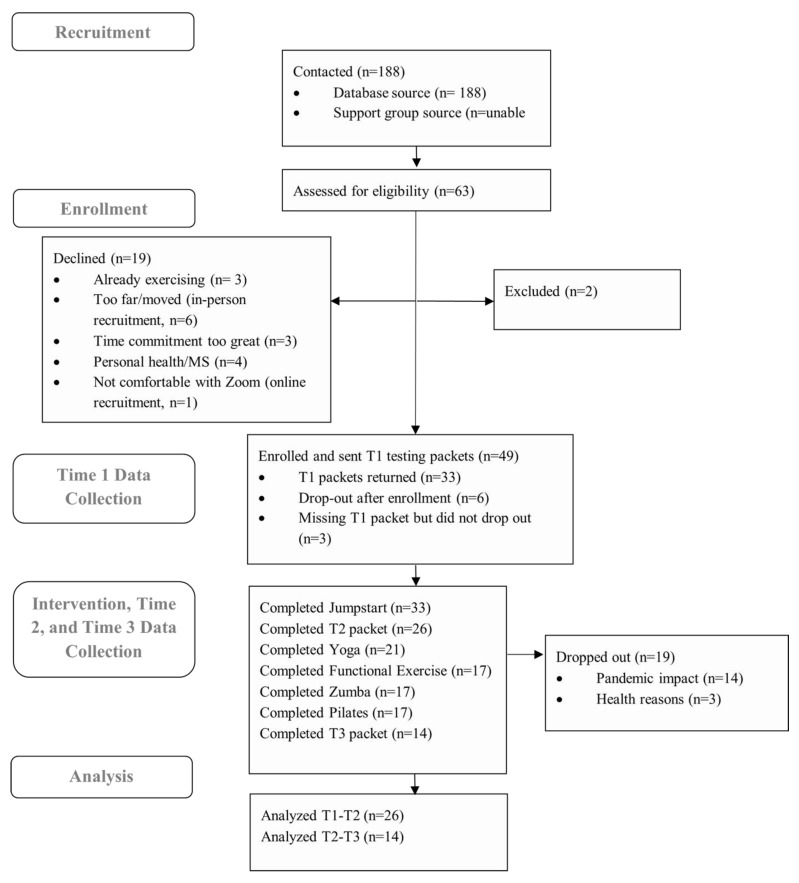
PRISMA Diagram.

**Figure 3 ijerph-20-06567-f003:**
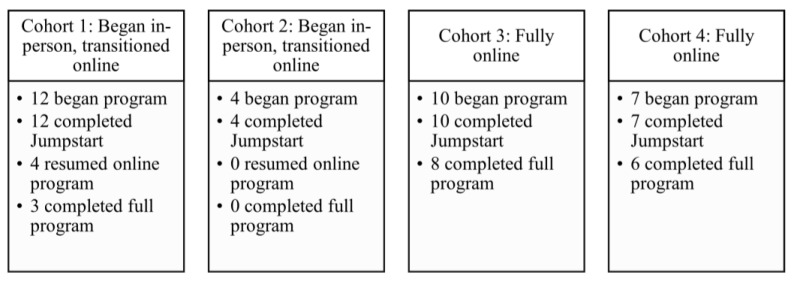
Cohort-by-cohort flow through MOVE MS.

**Table 1 ijerph-20-06567-t001:** Participant Demographics.

	Mean/N	SD/%
**Gender**		
Female	25	75.8
Male	7	21.2
Other	1	3.0
**Age**	55.2	1.7
Employment		
Employed	12	36.4
Unemployed	21	63.6
**Marital Status**		
Married	20	60.6
Never married	6	18.2
Divorced/separated	4	12.1
Widow/widower	3	9.1
**Race**		
Black or African American	3	9.1
White	29	87.9
Latino/a	1	3.0
**Education**		
High School Graduate	8	24.2
1–3 Years of College	8	24.2
College/University Graduate	11	33.3
Master’s Degree	3	9.1
PhD or Equivalent	3	9.1
**Income**		
<$25,000/year	5	16.7
$25,001–$45,000/year	8	26.7
$45,001–$65,000/year	4	13.4
>$65,001/year	13	43.3
**Type of MS**		
RRMS	25	78.1
PPMS	2	6.3
SPMS	4	12.5
Benign MS	1	3.1
**Number of years with MS**	15.8	9.7
PDDS		
0–2	18	54.5
3–5	10	30.3
6–8	5	15.1

RRMS = Relapsing remitting multiple sclerosis, PPMS = Primary progressive multiple sclerosis, SPMS = Secondary progressive multiple sclerosis, PDDS = Patient-Determined Disease Steps.

**Table 2 ijerph-20-06567-t002:** Feasibility results.

Metric	Monitoring and Assessment Strategy	Results
**Process: Critical program processes**	a. Recruitment and eligibility rates. b. Adherence, retention, compliance (attendance), and attrition rates.	a. **Recruitment rate**: 36 enrolled in program/188 contacted = 19.1%. **Eligibility rate**: 61 eligible/63 screened = 96.8%. b. **Adherence rate**: 17 completed intervention/33 began intervention = 52.0%. **Retention rate**: 17 completed intervention/33 began intervention = 52.0%. **Attendance**: 67% cohort 1, 75% cohort 2, 86% cohort 3, 79% cohort 4. **Attrition rate**: 22 did not complete T3 or dropped out/33 began the intervention = 66.7%.
**Resource: Time and monetary resource needs**	c. Communication with participants. d. Staff training time. e. Monetary costs of the intervention.	c. **Communication time**: = 1661 min. d. **Staff training time**: Instructor and research assistant training time = 850 min, e. **Monetary resources**: In-person to online cohorts = 1760.36 USD. Online only cohorts = 3701.80 USD.
**Management: Data management needs, intervention fidelity**	f. Staff time requirements for data collection, data entry, and checking. g. Missing data items. h. Intervention fidelity.	f. **Staff time needs for data collection**: = 1674 min. **Staff time needs for data entry and checking**: 1861 min. g. **Missing data items**: 50 missing items/13,486 total items = 99.6% complete h. **Jumpstart Fidelity**: Cohorts 1 and 2 completed as planned, Cohorts 3 and 4 completed virtually with same content. **Discover Fidelity**: Cohorts 1–4 received all 4 Discover modules in the order proposed, virtual instead of in person.
**Scientific: Participant burden, safety, and efficacy**	i. AEs, SAEs and clinical emergencies. j. Participant burden and satisfaction. k. Treatment effect.	i. **Adverse events**: 1 adverse event in 1 participant (possibly unrelated to the study), muscle soreness. **Serious adverse events**: none reported. j. **Participant burden: Mean time to complete questionnaires**: T1: 37.44 min, T2: 35.12 min, T2.5: 29 min, T3: 33.92 min. **Mean time to complete interview**: 65.42 min. **Participant Satisfaction**: Feedback summaries provided in supplementary material. k. **Primary scientific outcome**: GLTEQ T1 mean = 14.2 (SD = 11.8), T2 mean = 16.2 (SD = 11.7), T3 mean = 16.6 (SD = 11.2), F-value = 0.53 (Partial Eta^2^ = 0.08). See all other patient reported outcome measures in supplementary material.

## Data Availability

The data presented in this study are available on request from the corresponding author. The data are not publicly available due to the privacy of the participants involved who have not disclosed their MS diagnosis.

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
