# Peer review of "Results of the MOVE MS Program: A Feasibility Study on Group Exercise for Individuals with Multiple Sclerosis"

_ijerph, 2023, doi:10.3390/ijerph20166567_

Round 1

Reviewer 1 Report

Title: Results of the MOVE MS program: A feasibility study on group exercise for individuals with multiple sclerosis.

This feasibility study evaluated the exercise participation and several secondary outcomes in a 7-month trial MOVE MS program (a weekly group exercise program of multiple sclerosis patients based on Social Cognitive Theory involving behavior change education, multiple exercise modalities, and seated instruction).

Main comments

In general, the manuscript is well-written. Be careful with journal’s abbreviations. Some specific comments are presented below.

Specific comments

0. Abstract

- Line 30: Only “Multiple sclerosis” and “Feasibility study” Keywords are MeSH terms. It would be appropriate to use all MeSH terms in the Keywords section.

1. Introduction

- Line 90: To justify this kind of long-term intervention, it would be interesting to add the conclusions of references about similar programs to MOVE MS in other neurological diseases as stroke, spinal cord injury, traumatic brain injury…

2. Materials and Methods

- Line 114: Specify the frequency (1 session per week).

- Lines 125-127: Detail between parenthesis the dynamics in Yoga, Pilates and Zumba® Gold like the functional exercises.

- Lines 200-209: Write a short description of each scientific outcome measuring questionnaire/scale used.

- Line 211: Write the assumption of data normal distribution or the test used to revise this aspect.

3. Results

- No comments.

4.Discussion

- Line 505: Before “Conclusions” it would be appropriate to write about future recommendations, for instance, start again a similar long-term essay without COVID-19 factor, and compared the results to see the influence of the COVID 19 pandemic in MOVE MS program.

5. Conclusions

- Line 506-510: “Given the challenges…including depression” is an example of future recommendation. Conclusions must be related to the specific aims in lines 105-109.

Appendix A. Semistructured Interview Questions

- No comments.

References

- Write DOI instead of APA webpages format when could be possible (except for Internet resources).

- Line 605: Write “Sport Sci Health” instead of “Sport Sciences for Health”.

 - Line 632: Introduce the year, volume (issue) number and pages range from this article.

- Line 639: Introduce the volume (issue) number and pages range from this article.

- Lines 641 and 663: Write “APAQ” instead of “Adapted Physical Activity Quarterly”.

- Line 685: Write “HAJC” instead of “The Health & Fitness Journal of Canada”.

- Line 711: Write “IJQHC” instead of “International Journal for Quality in Health Care”.

Reviewer 2 Report

Here, the authors describe a structured intervention to improve physical activity among people with MS. This program, MOVE MS, endeavoured to improve long-term engagement with physical activity through a combination of instruction in exercise methods and behaviour change. This feasibility study of the MOVE MS program assessed changes in physical activity over 7 months follow-up, as well as secondary outcomes including MS symptoms, self-efficacy, depression, anxiety, disability, and QoL. The study started with 33 participants, of which 17 completed follow-up. The authors report nonsignificant effects on outcomes, including physical activity  participation. Qualitative interview of 16 participants completed the program found 5 themes relating to the program and feedback. The authors conclude the MOVE MS program may be a feasible intervention to improve physical activity in people with MS.

This is a comprehensive report, comprising both quantitative and qualitative analyses. That it is based on a rather small number of participants, only 17 completing the program and included in the quant analyses and only 16 in the qual, the results should be interpreted with caution. These results are suggestive of a potential benefit on maintaining physical activity, with some nonsignificant increases in physical activity by GLTEQ seen, and significant increases in perceived ability to engage in physical activity in the ESES. The effects on PROMs and other outcomes, however, are rather suspect with this sample size, so the authors should consider reducing the focus on these, or perhaps omitting from results altogether.

Please reduce the information provided in Table 2 to the most salient and relevant information. While much of this is of interest internally, particularly in the authors’ presumed plans for a follow-on study, they are less relevant to others. For instance, I’m thinking Management could be markedly reduced or omitted, time to complete Q’s in Scientific omitted.

Similarly, the results regarding program feedback I think might be moved to supplementary data.

The authors’ contentions regarding disability identity and its relevance to physical activity are relevant but do rather ignore the fact that disability is still an outcome worth reducing. While persons living with disability should not be shamed or otherwise stigmatised for this state, and those living with irreversible disability counselled on how to come to terms with this, disability is still an outcome that we can seek to at least halt the progression of, if not reverse it and improve function. Physical activity can be a means of pursuing these aims, through strengthening of muscles, both central and accessory, and also potentially through induction of anti-inflammatory processes which may directly impact upon the pathophysiology in MS. This also has the benefit of giving patients with MS a sense of control over their disease through their direct action. There is a need to endeavour to portray physical activity in a positive light that may have benefits on disability, among other outcomes in MS, and this can be done. While in some cases this delivery may not be done well, this is not cause to throw out the utility of physical activity with a goal to improve disability outcomes altogether. Thus, I might hope you could temper the negative framing of physical activity aiming to improve disability in the Intro, and speak to how it can be proposed to patients with different aims, reflecting the individual goals and situations of patients. That said, I do like the framing used in your program, namely “exercise does not serve to fix your MS but as a resource for overall well-being.”

Please consider changing terminology from exercise to physical activity. Exercise is potentially limiting and may suggest to patients that they have to engage in particular, structured formed of physical activity when really any kind of physical activity is worth engaging in, compared to doing nothing at all.

In the Abstract, please specify that the intervention is delivered weekly. Also, please provide some measures of association for the primary outcome and specify what of the secondary outcomes effects were seen.

Round 2

Reviewer 2 Report

My thanks to the authors for their thorough responses to my comments. I am happy with this revised manuscript.